# Overexpression of a Ramie (*Boehmaeria nivea* L. Gaud) Group I *WRKY* Gene, *BnWRKY49*, Increases Drought Resistance in A*rabidopsis thaliana*

**DOI:** 10.3390/plants13030379

**Published:** 2024-01-27

**Authors:** Yaning Bao, Yifei Zou, Xia An, Yiwen Liao, Lunjin Dai, Lijun Liu, Dingxiang Peng, Xing Huang, Bo Wang

**Affiliations:** 1Guizhou Key Laboratory for Tobacco Quality Research, College of Tobacco Science, Guizhou University, Guiyang 550025, China; ynbao@gzu.edu.cn; 2MOA Key Laboratory of Crop Ecophysiology and Farming System in the Middle Reaches of the Yangtze River, College of Plant Science and Technology, Huazhong Agricultural University, Wuhan 430070, China; lywen2009@126.com (Y.L.); mr__monkey@outlook.com (L.D.); liulijun@mail.hzau.edu.cn (L.L.); pdxiang@mail.hzau.edu.cn (D.P.); 3Rapeseed Research Institute, Guizhou Academy of Agricultural Sciences, Guiyang 550008, China; zyfinhzau@163.com; 4Zhejiang Xiaoshan Institute of Cotton & Bast Fiber Crops, Zhejiang Institute of Landscape Plants and Flowers, Zhejiang Academy of Agricultural Sciences, Hangzhou 311251, China; anxia@zaas.ac.cn; 5Environment and Plant Protection Institute, Chinese Academy of Tropical Agricultural Sciences, Haikou 571101, China

**Keywords:** ramie, WRKY transcription factors, drought, salt, stomatal aperture

## Abstract

Plants face multiple stresses in their natural habitats. *WRKY* transcription factors (TFs) play an important regulatory role in plant stress signaling, regulating the expression of multiple stress-related genes to improve plant stress resistance. In this study, we analyzed the expression profiles of 25 *BnWRKY* genes in three stages of ramie growth (the seedling stage, the rapid-growth stage, and the fiber maturity stage) and response to abiotic stress through qRT-PCR. The results indicated that 25 *BnWRKY* genes play a role in different growth stages of ramie and were induced by salt and drought stress in the root and leaf. We selected *BnWRKY49* as a candidate gene for overexpression in Arabidopsis. BnWRKY49 was localized in the nucleus. Overexpression of *BnWRKY49* affected root elongation under drought and salt stress at the Arabidopsis seedling stage and exhibited increased tolerance to drought stress. Further research found that *BnWRKY49*-overexpressing lines showed decreased stomatal size and increased cuticular wax deposition under drought compared with wild type (WT). Antioxidant enzyme activities of SOD, POD, and CAT were higher in the *BnWRKY49*-overexpressing lines than the WT. These findings suggested that the *BnWRKY49* gene played an important role in drought stress tolerance in Arabidopsis and laid the foundation for further research on the functional analysis of the *BnWRKYs* in ramie.

## 1. Introduction

The harsh environmental conditions such as drought, salinity, cold, and heat can have a devastating impact on plant growth and yield under field conditions [1], becoming one of the important factors restricting agricultural development. To cope with abiotic stress, plants undergo extensive molecular reprogramming at the transcriptional and post-transcriptional levels to enhance their adaptability to the environment. During this process, TFs play an important role [2]. *WRKY*-TFs are involved in plant growth and development and play important roles in response to abiotic and biotic stresses [3,4]. *WRKY* genes are divided into three groups (I, II, and III) according to the number of WRKY domains and the pattern of zinc-finger motifs. Members of group I usually have two WRKY domains containing a single C2H2 zinc-finger motif. Group II members contain a WRKY domain and a C2H2 zinc-finger motif. Members of group III have a WRKY domain, but they have different zinc-finger motifs, and the structure is C2HC [5]. At present, *WRKY* TFs have been identified at the genomic and transcriptome levels in 234 species, including lower plants, higher plants, protozoa, etc. [6]. The expression profiles of *AtWRKY25*, *AtWRKY26*, and *AtWRKY33* genes under abiotic stress were analyzed using Northern blot analysis, they responded to stresses including heat, NaCl, abscisic acid (ABA), and osmotic stress [7]. Recent research has shown that *AtWRKY20*, as a member of the WRKY I group, is a sucrose-enhanced TF that plays a role upstream of sucrose-responsive genes. Thereby, *AtWRKY20* affects various developmental and metabolic processes in plants [8]. In the phloem of plants infected with *Begomovirus*, *AtWRKY20* could be bound to the begomoviral βC1 protein. This viral hijacking of *AtWRKY20* spatiotemporally redeployed plant chemical immunity within the leaf [9]. *OsWRKY53* was induced by abscisic acid (ABA), jasmonate (JA), salt, drought, darkness, and low temperatures but inhibited by gibberellic acid (GA) and salicylic acid (SA) [10]. The *WRKY*-TFs in various woody plants were widely involved in responses to biotic and abiotic stresses, as well as in plant growth and development [11]. *PeWRKY41* belongs to Class III *WRKY* and interacts with casein kinase. Overexpression of *PeWRKY41* improved the insect resistance and salt tolerance of transgenic poplar [12]. Drought stress leads to osmotic and oxidative stresses due to cellular dehydration, ROS production is mainly regulated by NADPH oxidase, and *RBOH* is involved as a major regulatory gene in the regulation of plant responses to adversity stress [13]. The prolonged exposure to ROS leads to impaired cellular function, affecting plant growth and development. To cope with the sudden increase in ROS, plants have adopted various strategies for ROS detoxification and maintenance of cellular redox status [14]. Antioxidant enzymes such as CAT (catalase), SOD (superoxide dismutase), POD (Peroxidase), and GPX (Glutathione peroxidase) regulate the cellular redox status [15]. Compared with the WT, *TaGPX1-D*-overexpressing plants exhibit enhanced tolerance to salinity and osmotic stress, increased antioxidant enzyme activity and proline, and decreased levels of H_2_O_2_ and MDA (Malondialdehyde) [16]. WRKY played a key role in defense responses by regulating ROS homeostasis. In addition, cuticular wax was deposited on the surface of plants to reduce water loss and resist drought stress. *CitWRKY28* promoted the accumulation of epidermal wax in Arabidopsis leaves [17].

Ramie, which is also known as “China grass”, originated in China and is a perennial herbaceous plant of the Urticaceae family [18]. Ramie leaves are a rich source of proteins, minerals, vitamins, dietary fiber, phenolics, and flavonoids [19]. The fiber made from stem bast is a good textile material. With the acceleration of urbanization and industrialization in China, the per capita arable land area is gradually decreasing, and cultivation of ramie on mountain slopes or uncultivated land has become an option. However, the growth of ramie is affected by drought stress. It has therefore become urgent to improve the drought tolerance of ramie. In a previous study, twelve TFs from the *AP2*, *MYB*, *NAC*, *HD-Zip*, *bHLH*, *Dof*, *C2H2L*, *ARF,* and *GRAS* gene families were further verified to exhibit activity in response to drought stress [20], and from a transcriptome database, eight *WRKY* genes with intact, conserved domains were screened [21]. In another study, 25 TFs from the *AP2*, *MYB*, *NAC*, *zinc-finger*, *bZIP,* and *WRKY* families were reported to be potentially associated with drought stress [22]. The *BnWRKY* gene family has been identified, and the response of 12 *BnWRKYs* to cadmium stress has been studied [23]. Although the function of most Arabidopsis *WRKY*-TFs is known, the related research on the *WRKY* gene family in ramie is still limited.

In this study, we selected 25 *BnWRKY* genes from three subgroups of *BnWRKY* genes and analyzed the transcription patterns of the *BnWRKY* genes at three stages of ramie growth (the seedling stage, the rapid-growth stage, and the fiber maturity stage) and in response to the abiotic stresses (salt and drought). We further investigated the stress tolerance provided by *BnWRKY49* in transgenic Arabidopsis. This study will help us understand the potential molecular response mechanism for drought stress and will provide assistance to the breeding of drought-resistant ramie.

## 2. Results

### 2.1. Expression Patterns of the 25 BnWRKY Genes in Ramie

The expression patterns of these genes in four tissues were examined, including leaf tissue and three parts of stem bark tissue (the top, middle, and bottom parts) (Figure 1). The three samples of stem bark represented the different fiber growth stages: from the top to bottom parts, the fiber cell gradually thickened. At the seedling stage, more than half of the 25 *BnWRKYs* showed similar transcript levels between the young stems and leaves, and most of the 25 *BnWRKYs* exhibited the higher relative expression in the stem bark. Nine genes (*BnWRKY1*, *4*, *9*, *17*, *21*, *26*, *28*, *30*, *43*) exhibited low expression in the stem bark of the rapid growth stage. At the fiber maturation stage, compared to the stem bark, most of the 25 *BnWRKY* genes had higher expression levels in the leaves.

To study whether the expression pattern of the 25 *BnWRKYs* responded to abiotic stresses, qRT-PCR analyses were performed on the roots and leaves of hydroponic ramie seedlings (Figure 1). The results showed that all genes intensively responded to NaCl stress except *BnWRKY45* and *BnWRKY7*. Among these genes, the expression of *BnWRKY49* was downregulated in the roots under stress, while the transcripts of the others were highly promoted. In contrast, the expression of a majority of salt-responsive *BnWRKYs* (16 out of a total of 25 genes) was downregulated in the leaves, while only four genes had a greater expression. With respect to PEG stress, the relative expression level of seven genes showed no significance in either the root or leaf tissue. However, eight *BnWRKYs* presented transcript accumulation in the roots, five of which were obviously downregulated. Additionally, the expression of thirteen *BnWRKYs* increased in the leaves, while only three genes (*BnWRKY2*, *17*, and *49*) exhibited distinctly reduced expression levels. These stress-responsive genes can provide a better understanding of the *BnWRKY* genes related to abiotic stress.

### 2.2. Identification and Subcellular Localization of BnWRKY49

The *BnWRKY49* gene consisted of six exons and five introns (Figure 2a). Homologs in six other species were found and used to construct the phylogenetic tree. BnWRKY49 was most closely related to XP_010097277.1 from *Morus_notabilis* (Figure 2b). BnWRKY49 and its homolog amino acid sequences contained two WRKYGQK motifs and two C2H2 zinc-finger structures (Figure 2c), it belonged to group I, and its domains were highly conserved. The prediction of the tertiary structure model of the BnWRKY49 protein showed that it contains five β-sheet structures (Figure 2d).

The coding sequence of *BnWRKY49* gene fused in frame with GFP under the control of the CaMV 35S promoter was shown in Figure 3a. In Figure 3b, the GFP signals of the BnWRKY49-GFP fusion protein were detected only in the nucleus, suggesting that the BnWRKY49-GFP fusion protein was targeted to the nucleus and possibly functions as a TF.

### 2.3. Overexpression of BnWRKY49 Affects Root Elongation under Drought and Salt Stress

*BnWRKY49*-overexpressing transgenic plants formed longer primary roots than the WT in cultures supplemented with PEG and NaCl, and the root lengths of the three transgenic lines were significantly longer than the WT (Figure 4a,b).

### 2.4. Overexpression of BnWRKY49 Enhances Drought Stress Tolerance in Arabidopsis

Three *BnWRKY49*-overexpressing lines (L1, L2, and L3) under drought stress were evaluated to investigate the function of the *BnWRKY49*. At 7 days of drought stress, both WT plants and overexpression lines were subjected to drought stress and grew slowly, there was no significant difference between them. After 14 days of drought stress, WT exhibited more severe water-loss-related symptoms compared with the overexpression lines. When the plants were rewatered 7 days later, most *BnWRKY49*-overexpressing lines recovered growth, whereas WT plants did not. Three overexpression lines showed significantly higher recovery rates (Figure 5a). Therefore, under drought stress, the overexpression lines showed better growth and higher survival rates than the WT. We also determined the antioxidant enzyme activities of leaves from plants treated with drought for 7 days. The POD enzyme activities in the leaves of three *BnWRKY49*-overexpressing lines were higher than that of WT, but the difference was not significant. The activities of CAT and SOD enzymes of three overexpression lines were significantly higher than in WT (Figure 5b–d). The antioxidant enzyme activities were higher in the *BnWRKY49*-overexpressing lines under drought stress. The above results indicated that overexpression of *BnWRKY49* in Arabidopsis could improve drought resistance.

### 2.5. BnWRKY49 Induced Stomatal Closure and Promoted Cuticular Wax Accumulation under Drought Stress

By phenotypic observation, the *BnWRKY49*-overexpressing lines showed increased tolerance to drought stress. From this, we speculated that the difference may be caused by the difference in stomatal closure under drought stress (Figure 6a). Compared with WT plants, the stomatal openings of transgenic plants were shorter and narrower, and the aspect ratio was larger (Figure 6b,c). We also analyzed the cuticular wax of the WT plants and transgenic plants under drought stress with a scanning electron microscope. The results showed that, compared with WT plants, the *BnWRKY49*-overexpressing lines had more wax crystals on the stem surface (Figure 6d).

## 3. Discussion

The WRKY TFs, as a widely studied plant defense response protein, play a crucial regulatory role in resisting abiotic stress. The most significant of the various biological functions that *WRKY* TFs have in plants involves being key players in response to drought and salt stress [24]. Recently, increasing numbers of studies have supported the essential roles of *WRKY* TFs.

Herein, we cloned a novel *WRKY* TF, which belonged to the family group I, *BnWRKY49*. In our study, the transient transformation of the pEGAD-35S::BnWRKY49:GFP fusion vector in tobacco leaves showed a strong GFP signal exclusively in the nucleus. BnWRKY49 was localized in the nucleus, consistent with the results of other reported WRKY-TF [25]. The expression pattern of *BnWRKY49* was similar under drought stress and salt stress treatments, with downregulation in both roots and leaves. However, we overexpressed *BnWRKY49* in Arabidopsis, which enhanced the drought resistance of the overexpressed lines. The dynamics of *BnWRKY* expression changed in cadmium stress from 0 to 48 h. At 48 h, *BnWRKY10, 27, 40, 44*, and *46* showed downregulation of expression in roots, and *BnWRKY10* was an upregulated expression that reached its highest value at 6 h of the Cd^2+^ stress [23]. We analyzed that after abiotic stress treatment, we might not be able to observe the upregulation of *BnWRKY49* expression only after 48 h of sampling.

Drought severely affects crop growth, development, and yield. *WRKYs* enhance the drought tolerance of plants. *GmWRKY27* positively regulates drought tolerance [26]. *TaWRKY1* and *TaWRKY33* overexpressed in Arabidopsis participated in drought tolerance via ABA signaling [27], and the *OsWRKY30* gene enhanced the drought tolerance of rice through MAPK signaling [28,29]. In our study, it was found that ectopic overexpression of *BnWRKY49* enhanced drought stress tolerance in Arabidopsis. At the molecular level, plants usually respond to abiotic stress with the same signal transduction pathways, leading to the accumulation of ROS and ABA [30]. CAT, SOD, and POD enzymes act as scavengers of ROS to protect cells when plants are exposed to stresses and produce excess ROS. Our results indicated that the *BnWRKY49*-overexpressing plants exhibited higher CAT, SOD, and POD activities than the WT plants under drought conditions. Drought and salt stresses can lead to dehydration of cells and can affect plant yields and spatial distribution [30]. During long-term evolution, stomatal development (including density, closure, aperture, and size) and cuticle adjustments were the main defense mechanisms against water loss under drought stress [31,32,33,34,35]. Many reports are consistent with our research findings. Heterologous overexpression of the *GhWRKY41* gene in tobacco led to tolerance to drought and salt stress by enhancing stomatal closure and modulating clearance scavenging of ROS [36]. *AtWRKY70* and *AtWRKY54* cooperate as negative regulators of stomatal closure [37]. The *SlWRKY81* gene plays a negative role in drought stress by suppressing stomatal closure mediated by guard cells and H_2_O_2_ [38]. Moreover, overexpression of *GsWRKY20* resulted in decreased stomatal density and water loss rates, suggesting that *GsWRKY20* may act as a major transcriptional regulator of cuticular wax biosynthesis and accumulation in response to drought [34]. Under drought treatment, *TaWRKY31*-overexpressing lines in Arabidopsis lead to a decrease in H_2_O_2_ and MDA levels, reducing stomatal opening. In addition, an increase in antioxidant enzyme activity, germination rate, and root length was observed in *TaWRKY31* transgenic Arabidopsis [25]. Overexpression of the *TaNCL2-A* gene in Arabidopsis showed significant cadmium tolerance, salt tolerance, and osmotic stress, with increased calcium accumulation, increased proline content, and antioxidant enzyme activities, and decreased oxidative stress-related molecules such as H_2_O_2_ and MDA [39].

However, little is known about the role of *BnWRKYs* in regulating drought response in ramie. The qRT-PCR results showed that the *BnPP2C1* and *BnPP2C26* genes responded to drought, high levels of salt, and ABA treatments, and there were a large number of stress-responsive elements in the promoter region, which can be used as candidate genes for drought and salt tolerance in ramie [40]. *BnbZIP2* gene overexpression in Arabidopsis showed high sensitivity to drought stress at the seed germination stage and higher tolerance to salinity stress than the WT [41]. The molecular mechanism research related to drought resistance in ramie is still limited [42]. In our study, the stomatal openings of the transgenic plants were longer and wider, with a smaller length/width ratio, and less cuticular wax was observed on the surfaces of WT plants compared with the transgenic plants. Thus, *BnWRKY49* may positively regulate the expression of genes involved in wax biosynthesis and may regulate stomatal opening. WRKY-TFs are involved in plant signaling pathways (e.g., phytohormones, MAPKs, ROS, Ca^2+^) and metabolic regulation associated with stress responses [43]. Recently, a novel mechanism of DPY1-SAPK6-mediated drought tolerance was revealed [44]. WRKY-mediated crosstalk between abiotic and biotic stress responses is a common element in plants. WRKYs have high potential applications in crop improvement in the future.

## 4. Materials and Methods

### 4.1. Plant Materials and Treatments

Ramie cultivar Huazhu No. 5 was cultivated at the Ramie Germplasm Resources Garden of Huazhong Agricultural University (Wuhan, China). We harvested leaves and stem bark in three growth stages of ramie, including the seedling stage, the rapid-growth stage, and fiber maturity stage. During the seedling stage, we took the young stem bark (YS) and leaves (YL). During the rapid-growth stage and the fiber maturity stage, we adopted the method of Chen et al. [45] for sampling. We took stem bark from the top, middle, and bottom positions of the stem, and young leaves, and named them TS, MS, BS, and L.

For stress treatments, we cut off the young shoot tips (approximately 15 cm) of Huazhu No. 5 from the Ramie Germplasm Resources Garden and cultivated them in the greenhouse (28 °C/16 h, light; 20 °C/8 h, dark). The young shoot tips were soaked in 0.1 g/L KMnO_4_ for 2 days and then transferred into a conical flask filled with tap water for rooting. After rooting, all plantlets were placed into half-strength Hoagland’s solution and grown for 20 days. Afterward, the plantlets were treated with 150 mM NaCl (representing salt stress), and the roots and leaves were sampled after 20 h for salinity stress. Other plantlets were cultured in 20% PEG 6000 (*w*/*v*) (representing drought stress), and the roots and leaves were sampled after 48 h for drought stress. Untreated plants served as controls, and all the treatments were carried out in three biological replicates. All samples were immediately placed in liquid nitrogen and then stored at −80 °C for RNA isolation. 

### 4.2. Total RNA Extraction and Quantitative Real-Time Reverse Transcription PCR (qRT-PCR) Analysis

We extracted the total RNA using the RNAPrep Pure plant kit (Tiangen Biotech, Beijing, China). The complementary DNA was synthesized by the GoScript reverse transcription system (Promega, Madison, WI, USA). qRT-PCR analysis was conducted via an iQ5 multicolor real-time PCR system (Bio-Rad, Hercules, CA, USA). Here we analyzed the expression of 25 *BnWRKY* genes from three subgroups [23]. The primers (Appendix A) applied to qRT-PCR were generated from the Primer3 website (http://primer3.ut.ee/, accessed on 1 June 2015) and were further evaluated by Oligo 7 (Molecular Biology Insights Inc., Cascade, CO, USA). The *EF1A* gene was selected as a reference gene. Three biological and technical replicates were performed, and the relative expression levels were calculated based on the comparative cycle threshold (2^−∆∆Ct^) values [46]. Heatmap Illustrator 1.0 was used to generate a comprehensive presentation of gene expression under the different tested conditions.

### 4.3. Phylogenetic Tree and Sequence Alignment Analyses of BnWRKY49

The young leaves of ramie cultivar Huazhu No. 5 were used for the extraction of genomic DNA with an HP Plant DNA Kit (Omega, Norcross, GA, USA). The cDNA was derived from Section 4.2. The *BnWRKY49* CDS sequence obtained from whole-genome analysis were used as templates [24]. We designed three pairs of primers by Oligo 7.0 to clone *BnWRKY49* (Appendix A), including genomic sequence and CDS sequence. The amplified products were A-T cloned into the pEASY-T5 Zero Vector and sequenced. The intron-exon structure of *BnWRKY49* analysis was carried out via GSDS 2.0 (http://gsds.gao-lab.org/, accessed on 15 January 2018). We compared the BnWRKY49 protein with homologous proteins from *Populus_trichocarpa* (Potri.001G361600.2), *Glycine*_*max* (Glyma.14G016200.1), *Oryza_sativa* (LOC_Os07g39480.1), *Zea_mays* (GRMZM2G130854_P01), *Morus_notabilis* (XP_010097277.1), and Arabidopsis (AT4G26640.2) to construct an evolutionary tree via the neighbor-joining (NJ) method using MEGA 7 software, and the parameter of bootstrap repetition was 1000. Multiple sequence alignments were performed using the online multiple sequence alignment of muscle (https://www.omicshare.com/tools/Home/Soft/musclecmp, accessed on 15 December 2023). The tertiary structure prediction model of the BnWRKY49 protein was predicted using the Phyre 2 (http://www.sbg.bio.ic.ac.uk/phyre2/html/page.cgi?id=index, accessed on 15 January 2018).

### 4.4. Subcellular Localization of BnWRKY49 Proteins

To determine the subcellular localization of BnWRKY49 proteins, *BnWRKY49:GFP* fusion genes controlled by the CaMV 35S promoter were generated. The fusion genes were then transferred into tobacco plants via *Agrobacterium*-mediated transformation and labeled. After the tobacco plants were incubated at room temperature in the dark for 48 h, the epidermal cells were detected by a laser scanning confocal microscope (Olympus FV1200, Tokyo, Japan) with green fluorescence excited with a 488 nm laser. The untreated tobacco leaves were used as negative controls to exclude autofluorescence [47]. Three biological replicates and the imaging data were processed using Photoshop CC software (Adobe Inc., San Jose, CA, USA).

### 4.5. Vector Construction and Transformation

Specific primers (Appendix A) were designed, and the cDNA was used as the template for the cloning of the *BnWRKY49* CDS sequence. The fragment was then ligated into the pMDC32 plant expression vector at the *KpnI* and *SacI* sites under the control of the 2× 35S promoter. The construct was transformed into *Agrobacterium tumefaciens* GV3101 by electroporation, which was then transformed into Arabidopsis (Col-0) using the floral-dip method [48]. Positive transformants on hygromycin (50 μg/mL) plates were selected. Total DNA was extracted using the cetyltrimethylammonium Ammonium Bromide (CTAB) method and confirmed by PCR using specific primers (Appendix A). Abiotic stress tolerance analysis was performed using homozygous T3-generation plants.

### 4.6. Analyses of Root Growth under Drought and Salt Stresses

The seeds of transgenic plants were placed in 1.5 mL EP tubes, after which 70% ethanol was added for 1 min. Afterward, 10% (*v*/*v*) NaClO was added for 8 min, and then the seeds were washed 5 times with water. Half-strength Murashige and Skoog (1/2 MS) medium containing the seeds were incubated in the dark at 4 °C for 3 d and then incubated under 16 h light/8 h dark conditions at 22 °C for 5 d. For root growth measurements, 5-day-old transgenic Arabidopsis and WT seedlings were transferred to 1/2 MS medium containing 125 mM NaCl. Due to PEG cannot be dissolved in agar, we adopted the protocol described by Verslues et al. [49]. The agar medium and PEG covering layer (excluding agar) were prepared separately, the final water potential of the PEG-infused plate having *ψ*w = −0.75 MPa. First, 1/2 MS agar medium was poured into the regular (100 mm diameter) plate, and then the PEG covering layer was placed on the solidified 1/2 MS agar medium. The volume ratio of the agar medium to PEG covering layer was 2:3, and it was left overnight (12–15 h) to allow PEG to diffuse into the agar. Before use, pour out the solution. The seedlings were then grown vertically. Imaging and root length measurements were performed after 7 d.

### 4.7. Drought Tolerance Assays

For drought tolerance assessments, we mixed the vermiculite and nutrient soil in a 1:1 ratio, filled the pot with the same weight, and absorbed water through the small holes at the bottom to completely moisten the soil. WT and three *BnWRKY49* transgenic Arabidopsis lines grown under normal watering for 18 days were subjected to drought treatment. The phenotypes were observed on the 7th and 14th days when watering was stopped. When WT Arabidopsis exhibited a severe dehydration phenotype, watering was resumed and the phenotype was observed 7 days after rewatering.

### 4.8. Measurements of the Antioxidant Enzyme Activities

Leaves were sampled from 7d drought-treated WT Arabidopsis and three *BnWRKY49* overexpression lines to measure the antioxidant enzyme activities. The peroxidase (POD) [50], superoxide dismutase (SOD), and catalase (CAT) [51] were measured according to the manufacturers’ instructions of the assay kit (Nanjing Jiancheng Bioengineering Institute, Nanjing, China). Three biological and technical replicates were performed.

### 4.9. Stomatal Aperture and Cuticular Wax

Leaves and stems during the same growth period and at the same position were sampled from 7d drought-treated WT Arabidopsis and three *BnWRKY49* overexpression lines, put into a buffer, and then subjected to a vacuum to allow the buffer to infiltrate the leaves. The stomata of the leaves and stems were viewed using scanning electron microscopy (SEM). For statistical analysis, the length and width of the stomata were measured by ImageJ software (National Institutes of Health, Bethesda, MD, USA) [52], and the stomatal length/width ratio was calculated and used as a measurement of stomatal closure [53]. The epicuticular waxes of the stem were also observed via SEM. Three biological replicates were performed.

### 4.10. Statistical Analysis

The data were analyzed by analysis of variance (ANOVA), and the means values were compared by *t*-tests using SPSS 11.5 (SPSS, Chicago, IL, USA). Figures of the bar charts were produced by Prism 7 (GraphPad Software Inc., San Diego, CA, USA).

## 5. Conclusions

In conclusion, the present study suggested that *BnWRKY49*, as a type I WRKY TF, played an important role in abiotic stress. The gene structure analysis and phylogenetic analysis revealed its conserved nature. The expression profiling suggested the involvement of BnWRKY49 proteins in plant development and abiotic stress responses. *BnWRKY49* showed increased root length in *BnWRKY49-*overexpressing Arabidopsis under NaCl and PEG treatment, as well as enhanced drought resistance by the stomatal closure and the density of surface wax. We speculated that it played a positive regulatory role in the resistance to drought stress. This study will provide ideas for ramie breeders to research and develop drought-resistant ramie varieties.

## Figures and Tables

**Figure 1 plants-13-00379-f001:**
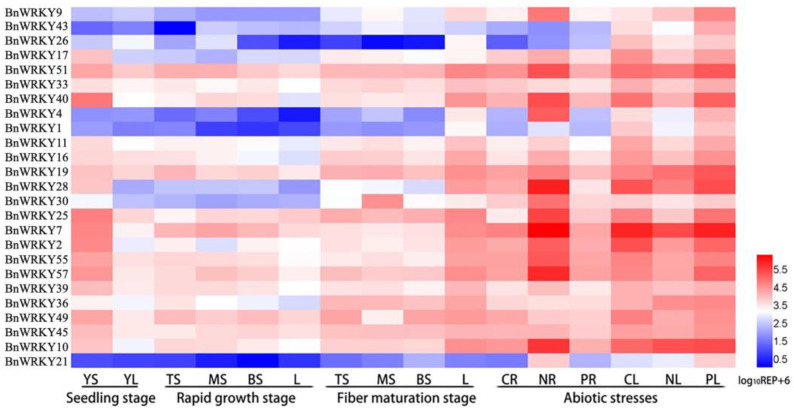
Expression pattern analysis of *BnWRKYs*. Relative expression of 25 *BnWRKYs* in the seedling stage (YS: stem bark; YL: leaves), rapid growth stage (TS: top of stem bark; MS: middle of stem bark; BS: bottom of stem bark; L: leaves), fiber maturation stage (TS: top of stem bark; MS: middle of stem bark; BS: bottom of stem bark; L: leaves), and under abiotic stress (CR, NR, and PR represent roots under control, NaCl, and PEG treatments, respectively; CL, NL, and PL represent leaves under control, NaCl, and PEG stresses, respectively).

**Figure 2 plants-13-00379-f002:**
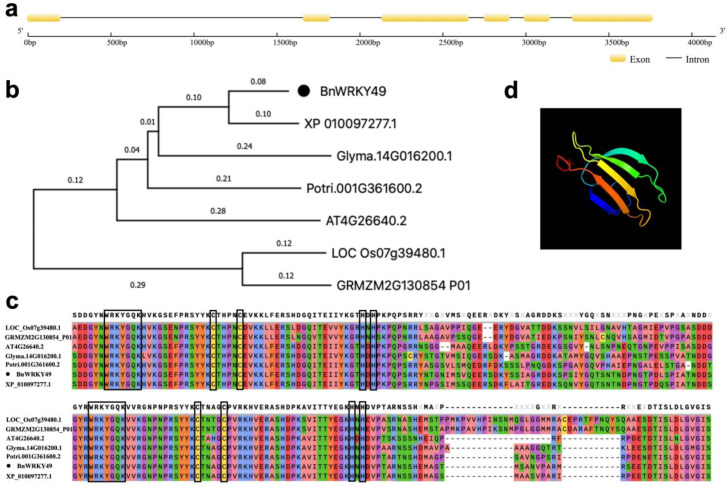
Bioinformatics alignment of BnWRKY49. (**a**) Gene structure of *BnWRKY49*. (**b**) Phylogenetic tree of BnWRKY49 and homologous genes in other species. (**c**) WRKY domains and conserved domains within proteins of the BnWRKY49 with homologous gene sequences in other species. Two WRKYGQK motifs and two C2H2 zinc-finger structures were labeled with black boxes. (**d**) Tertiary structure prediction model of the BnWRKY49 protein.

**Figure 3 plants-13-00379-f003:**
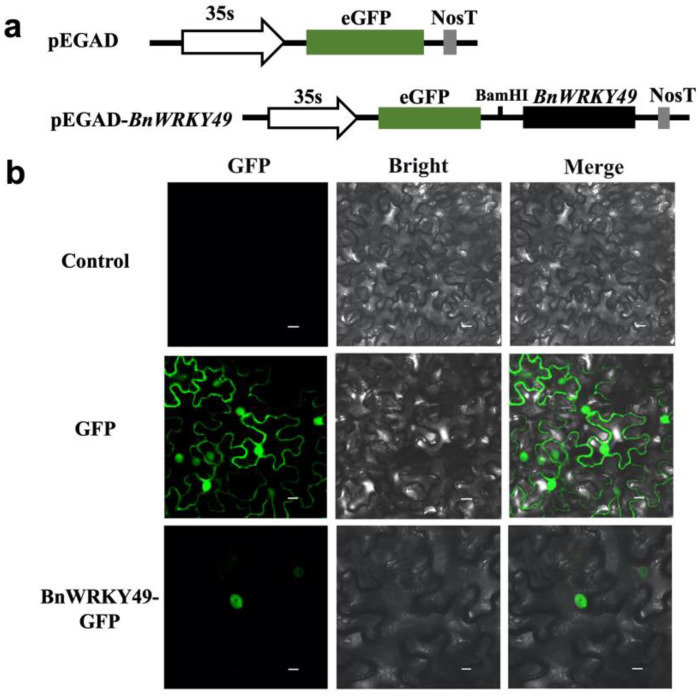
Subcellular localization of four BnWRKY49 proteins in tobacco leaves. (**a**) The structures of pEGAD-35S::BnWRKY49:GFP fusion vector and pEGAD-35S::GFP. (**b**) Green fluorescence images, bright field images, and the merge images overlay of bright and GFP were captured. The untreated tobacco leaves were used as a negative control to exclude autofluorescence. Tobacco leaves that were treated with an empty pEGAD vector were named GFP as a positive control. Bars, 20 μm.

**Figure 4 plants-13-00379-f004:**
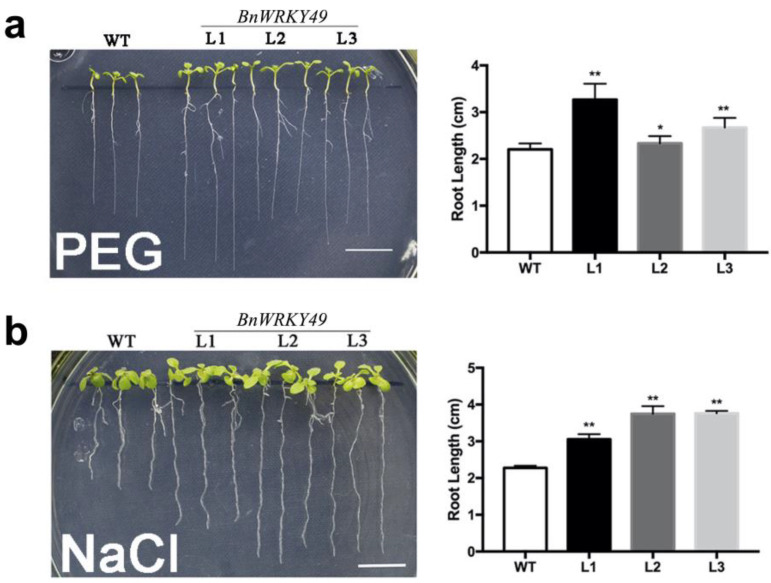
Assay of the root length of *BnWRKY49* overexpression Arabidopsis seedlings under PEG and NaCl stress treatment. (**a**) The root growth of Arabidopsis seedlings in the final water potential of the PEG-infused plate (−0.75 MPa) for 7 d. (**b**) The root growth of Arabidopsis seedlings under 125 mM NaCl stress for 7 d. From left to right are WT and three *BnWRKY49* overexpressing lines L1, L2, and L3, respectively. The error bars represent the mean (SD) of ten biological replicates (* *p* < 0.05, ** *p* < 0.01).

**Figure 5 plants-13-00379-f005:**
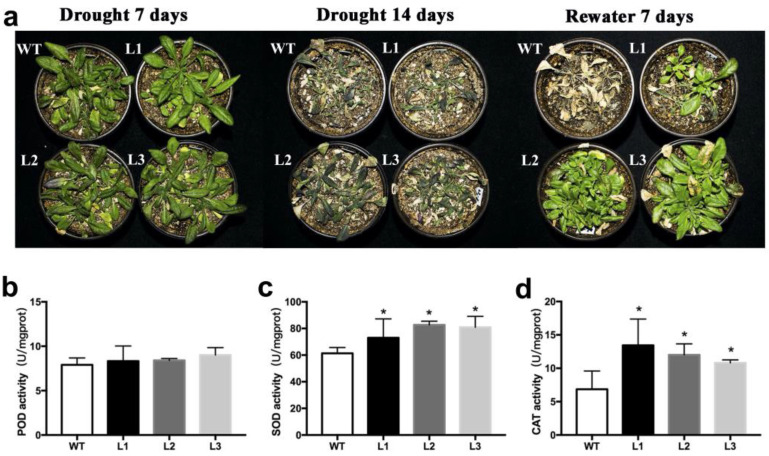
The phenotype of *BnWRKY49*-overexpressing Arabidopsis that were exposed to drought stress. (**a**) The phenotype of WT and three *BnWRKY49*-overexpressing lines (L1, L2, L3) exposed to drought stress for 7 and 14 days and rewatered for 7 days. (**b**) POD activity. (**c**) SOD activity. (**d**) CAT activity. The error bars represent the mean (SD) of three biological replicates (* *p* < 0.05).

**Figure 6 plants-13-00379-f006:**
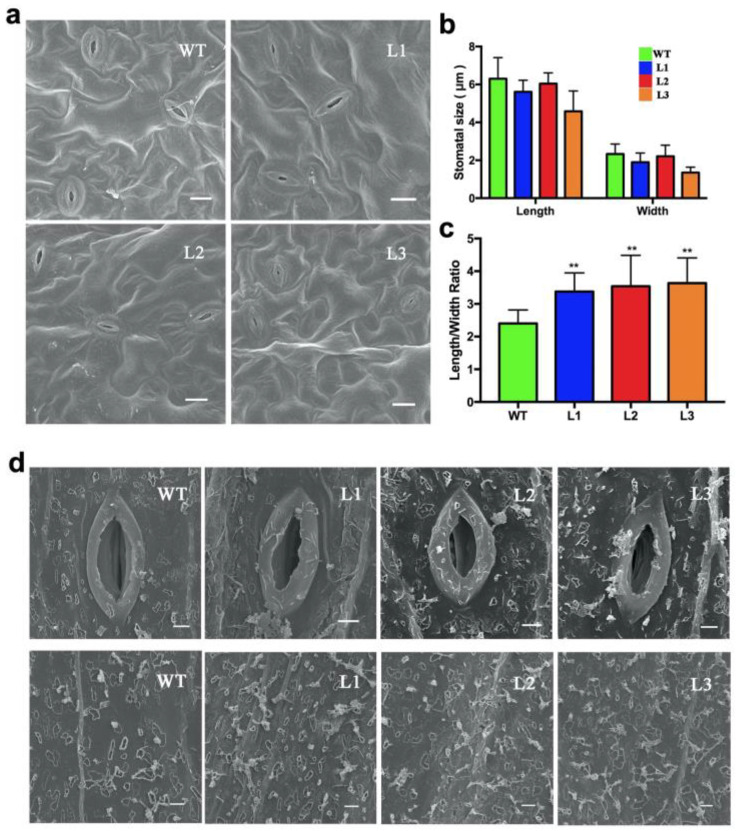
Stomatal aperture and epicuticular wax crystals of *BnWRKY49-*overexpressi*ng* Arabidopsis under drought stress. (**a**) Stomatal aperture of the WT plant and transgenic plant leaves. The stomata were observed using SEM; bars, 10 μm. (**b**) The size of the stomata. (**c**) The length/width ratio of the stomata. The error bars represent the mean (SD) of at least three biological replicates (** *p* < 0.01). (**d**) The picture above shows SEM images of the stomatal aperture and epicuticular wax crystals on the WT plant and transgenic plant stems that were exposed to drought stress. Bars, 2 μm.

## Data Availability

The data that support the findings of this study are available from the corresponding author, upon reasonable request.

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
