# Peer review of "Overexpression of a Ramie (Boehmaeria nivea L. Gaud) Group I WRKY Gene, BnWRKY49, Increases Drought Resistance in Arabidopsis thaliana"

_plants, 2024, doi:10.3390/plants13030379_

Round 1
Reviewer 1 Report
Comments and Suggestions for Authors
The Ms “ Molecular Clone and expression analysis of the WRKY transcription factor family in ramie (Boehmaeria nivea L. Gaud) and Overexpression of BnWRKY49 increases drought tolerance in Arabidopsis ‘ needs to be thoroughly revised, including from title to the conclusion.
What is meaning of molecular clone in the title? Authors even do not have the idea how to write the title.
First of all the complete methods should be revised. Why genes have been cloned from the genomic DNA, why not from cDNA, or from both?
How gene structure could be analysed only with the clone from genomic DNA?
Which DNA was used for the vector construction for localization and transformation? Nothing is clear.
How primers were designed for cloning? What was template for primer desgning, where authors got sequence infor for primer designing? Author should first read some recent articles from good journals like PMID: 37879539 and ‘Decoding the features and potential roles of respiratory burst oxidase homologs in bread wheat’ recently published in reputed journals, how to write the methodology. Further, for drought signalling authors should read some recent Ms like’ DPY1 as an osmosensor for drought signaling’ and other for discussion. The drought tolerance has been achieved by expressing numerous other genes, some of the recent studies, like expression of TaGPX1-D, TaNCL2-A can be discussed in the discussion section
Other minor changes-
1. In line no. 29 there is no need to write Arabidopsis in brackets.
2. In line no. 62 and 63 give the complete name of JA, SA, and GA.
3. In Figure 1 legends, give the complete information of YS, YL, TS, MS, BS, L, CR, NR, PR, CL, NL, PL, otherwise it is little difficult to understand.
4. In line no. 134 correct the spelling of construct.
5. It is necessary to read over the entire manuscript to correct grammatical errors and improve English.
6. The image quality is poor. It is advised to provide figures with higher resolutions.
7. Line no. 158 and 159 are suggested to be reframed.
8. The results of enzyme activity of WT and transgenic lines are not well written. Describes the results in detail.
9. Discussion part should be in more detail.
10. In section 4.7 of material and methods, you did not mention the percentage of PEG. Give the PEG %age you have used in your transgenic analysis.
Comments on the Quality of English Language
Extensive editing of English language required
Author Response
Dear Reviewer,
Thank you very much for taking the time to review this manuscript entitled “Molecular Clone and expression analysis of the WRKY transcription factor family in ramie (Boehmaeria nivea L. Gaud) and Overexpression of BnWRKY49 increases drought tolerance in Arabidopsis” (Manuscript Number: plants-2830052). We would like to express our sincere thanks to the reviewers for the constructive and positive comments. Here we did not list the changes but marked the revisions in red.
Comments 1: What is meaning of molecular clone in the title? Authors even do not have the idea how to write the title.
Response 1: We revised the title as suggested. Thanks for your useful suggestion.
Comments 2: First of all the complete methods should be revised. Why genes have been cloned from the genomic DNA, why not from cDNA, or from both?
Response 2: Based on the structure of the manuscript, we removed the part of the 25 BnWRKYs clone. We only kept the description of BnWRKY49 clone in 4.3., and the gene clone both from genomic DNA and cDNA. Thanks for your useful suggestion.
Comments 3: How gene structure could be analysed only with the clone from genomic DNA?
Response 3: Sorry for our incorrect description. We analyzed the gene structure with the CDS sequence and the genomic sequence. We revised the methods. Thanks for your useful suggestion.
Comments 4: Which DNA was used for the vector construction for localization and transformation? Nothing is clear.
Response 4: We revised the methods as suggested. Thanks for your useful suggestion.
Comments 5: How primers were designed for cloning? What was template for primer desgning, where authors got sequence infor for primer designing?
Response 5: We designed primers based on the cDNA sequences, and cloned the genomic sequences in the DNA. We revised the methods. Thanks for your useful suggestion.
Comments 6: Author should first read some recent articles from good journals like PMID: 37879539 and ‘Decoding the features and potential roles of respiratory burst oxidase homologs in bread wheat’ recently published in reputed journals, how to write the methodology. Further, for drought signalling authors should read some recent Ms like’ DPY1 as an osmosensor for drought signaling’ and other for discussion. The drought tolerance has been achieved by expressing numerous other genes, some of the recent studies, like expression of TaGPX1-D, TaNCL2-A can be discussed in the discussion section
Response 6: We further modified the methodology according to the Ref. We rewrote the discussion of the manuscript as suggested. Thanks for your useful suggestion.
Comments 7: In line no. 29 there is no need to write Arabidopsis in brackets.
Response 7: Revised as suggested. Thanks for your useful suggestion.
Comments 8: In line no. 62 and 63 give the complete name of JA, SA, and GA.
Response 8: We are sorry for our incorrect writing, and rewrite these sentences. Thanks for your useful suggestion.
Comments 9: In Figure 1 legends, give the complete information of YS, YL, TS, MS, BS, L, CR, NR, PR, CL, NL, PL, otherwise it is little difficult to understand.
Response 9: We revised the legend of Fig.1 as suggested. In addition, we also revised the Materials and Methods to make it easier for readers to understand. Thanks for your useful suggestion.
Comments 10: In line no. 134 correct the spelling of construct.
Response 10: We are sorry for our incorrect writing, and rewrite these sentences.
Comments 11: It is necessary to read over the entire manuscript to correct grammatical errors and improve English.
Response 11: Revised as suggested. Thanks for your useful suggestion.
Comments 12: The image quality is poor. It is advised to provide figures with higher resolutions.
Response 12: We have replaced all the images with better quality as suggested. Thanks for your useful suggestion.
Comments 13: Line no. 158 and 159 are suggested to be reframed.
Response 13: Revised as suggested. Thanks for your useful suggestion.
Comments 14: The results of enzyme activity of WT and transgenic lines are not well written. Describes the results in detail.
Response 14: Revised as suggested. Thanks for your useful suggestion.
Comments 15: Discussion part should be in more detail.
Response 15: We improved the Discussion as suggested. Thanks for your useful suggestion.
Comments 16: In section 4.7 of material and methods, you did not mention the percentage of PEG. Give the PEG %age you have used in your transgenic analysis.
Response 16: We have provided a detailed description of this section and highlighted it the manuscript. Thanks for your useful suggestion.
Thank you very much for your comments and suggestions. We hope that the revision is acceptable, and I look forward to hearing from you soon.
Sincerely,
Yaning Bao
Reviewer 2 Report
Comments and Suggestions for Authors
a few notes added that could be changed

a few notes to change
Author Response
Dear Reviewer,
Thank you very much for taking the time to review this manuscript entitled “Molecular Clone and expression analysis of the WRKY transcription factor family in ramie (Boehmaeria nivea L. Gaud) and Overexpression of BnWRKY49 increases drought tolerance in Arabidopsis” (Manuscript Number: plants-2830052). We gratefully appreciate your valuable comment. we have revised it according to your ideas, and here we did not list the changes but marked the revisions in red.
We hope that the revision is acceptable, and I look forward to hearing from you soon.
Sincerely,
Yaning Bao
Reviewer 3 Report
Comments and Suggestions for Authors
Comments and Suggestions for Authors
Dear Author,
I have an honor to review the manuscript entitled “Molecular Clone and expression analysis of the WRKY transcription factor family in ramie (Boehmaeria nivea L. Gaud) and Overexpression of BnWRKY49 increases drought tolerance in Arabidopsis” a research article submitted to MDPI Journal, Plants. Authors of this manuscript cloned and characterized 25 WRKY transcription factors (TFs) in ramie. They found that those genes play roles in different growth stages of ramie. Further, overexpressed the BnWRKY49 gene in Arabidopsis that increased tolerance to drought and salt at the seedling stage. Overall, the experiments are performed well and the results are convincing. Thus, the presented results takes up an important topic consistent with the profile of the Journal.
-However, even, manuscript is well organized and well described of the conception, I have some suggestions, which might improve the manuscript to make important to the wider audience.
-Few suggestions I have mentioned in the main text pdf file. Please check
-Some comments are as below
-Firmed aim of the study that should be underlined precisely and simultaneously and highlight why the TF analysis is important to study in ramie.
-There are many places where grammar can be improved. I suggest a careful revision by a professional language editing service. Extensive editing of English language and style is required. I've just noted a few here.
Abstract: -Good organization with results order.
L31-34; Rephrase this sentence
-“increased tolerance to drought”-----rephrase sentences and limit using similar wording and redundancy
Introduction:
-Introduction is not straightforward relating to results. Many unnecessary description also highlighted
L87; “But the research on the WRKY gene family in ramie is still limited”. Even limited, but you did not mention any in the introduction. That is limitation of your study
Ref. A total of 60 WRKY family genes of ramie were identified in the ramie. https://doi.org/10.3389/fpls.2022.812988
-Introduction should be more informative and sequential including some more specific findings referencing recent publications. Rationale to be elucidated for the purpose of the study. Write more information similar to your work.
-some relevant implication about methodology should be presented in the Introduction, which lacks here.
2. Results
-Why 25 genes were selected to clone? Give rationale
Fig. 1; provide the program used
2.3. Identification and Subcellular localization of BnWRKY49; why suddenly analyzed only this gene. Why not any other. Give rationale
Fig. 4; there should be a, b, c, d. what is L1, L2, L3? Text description is very limited. Legend also need improvement. SE indication in the figure does not meet the ±. It is only +. Same for Fig. 5B also.
Figure 5A. WT, L1, L2, L3 indication in the fig. is not appropriate. Point correctly. Either at upper or lower of the fig.
3. Discussion
This is somewhat weak discussion for the journal Plants. I suggest, some improvement having results comparing with recent article in ramie as well as Arabidopsis, even other plants related to WRKY functions, structures, and others.
4. Methods
-need some discussion about cultivation
-It is important to indicate the time of database browsed
L272; Total RNA was extracted from where?
L328; Leaves from where?
-Enzymatic assay need description of references
Most analysis was from Arabidopsis, not from ramie
5. Conclusions
Conclusion is too scanty. Does not represent the title and abstract. However, have reflection of partial results.

Moderate editing of English language required
Author Response
Dear Reviewer,
Thank you very much for taking the time to review this manuscript entitled “Molecular Clone and expression analysis of the WRKY transcription factor family in ramie (Boehmaeria nivea L. Gaud) and Overexpression of BnWRKY49 increases drought tolerance in Arabidopsis” (Manuscript Number: plants-2830052).Thank you very much for your comments and suggestions, here we did not list the changes but marked the revisions in red.
Comments 1: -Firmed aim of the study that should be underlined precisely and simultaneously and highlight why the TF analysis is important to study in ramie.
Response 1: Revised as suggested. Thanks for your useful suggestion.
Comments 2: -There are many places where grammar can be improved. I suggest a careful revision by a professional language editing service. Extensive editing of English language and style is required. I've just noted a few here.
Response 2: We gratefully appreciate for your valuable comment. we have done it according to your ideas.
Comments 3: L31-34; Rephrase this sentence
-“increased tolerance to drought”-----rephrase sentences and limit using similar wording and redundancy
Response 3: Revised as suggested. Thanks for your useful suggestion.
Comments 4: -Introduction is not straightforward relating to results. Many unnecessary description also highlighted
Response 4: Revised as suggested. Thanks for your useful suggestion.
Comments 5: L87; “But the research on the WRKY gene family in ramie is still limited”. Even limited, but you did not mention any in the introduction. That is limitation of your study
Ref. A total of 60 WRKY family genes of ramie were identified in the ramie.https://doi.org/10.3389/fpls.2022.812988
Response 5: In this article, the author identified 60 WRKY genes and studied the response of 12 BnWRKYs to cadmium stress, lack of in-depth functional validation of BnWRKY gene. We added the research on drought resistance in ramie to the manuscript, and revised the introduction and discussion. Thanks for your useful suggestion.
Comments 6: -Introduction should be more informative and sequential including some more specific findings referencing recent publications. Rationale to be elucidated for the purpose of the study. Write more information similar to your work.
Response 6: We revised the introduction as suggested. Thanks for your useful suggestion.
Comments 7: -some relevant implication about methodology should be presented in the Introduction, which lacks here.
Response 7: We revised the introduction as suggested. Thanks for your useful suggestion.
Comments 8: -Why 25 genes were selected to clone? Give rationale
Response 8: In this article (Ref. A total of 60 WRKY family genes of ramie were identified in the ramie. https://doi.org/10.3389/fpls.2022.812988), a total of 60 BnWRKY genes were identified, and the expression of 12 genes was analyzed under cadmium stress. Because 60 genes were too many, we randomly selected 25 genes from the three WRKY subgroups for expression analysis under abiotic stresses. We revised it as suggested. Thanks for your useful suggestion.
Comments 9: Fig. 1; provide the program used
Response 9: Figure 1 was produced by Heatmap Illustrator 1.0. We mentioned it in the 4.2. Thanks for your useful suggestion.
Comments 10: 2.3. Identification and Subcellular localization of BnWRKY49; why suddenly analyzed only this gene. Why not any other. Give rationale
Response 10: Among these 25 genes, there are many genes worthy of functional study. Our selection criteria are based on the results of qRT-PCR. In fact, we have selected more than one gene for functional validation, but in this manuscript, we only introduced the function of the BnWRKY49 gene. The expression level of BnWRKY49 gene was significantly changed in roots and leaves under salt and drought stress. Therefore, we speculate that BnWRKY49 have related functions. We have supplemented this section in the manuscript. Thanks for your useful comments.
Comments 11: Fig. 4; there should be a, b, c, d. what is L1, L2, L3? Text description is very limited. Legend also need improvement. SE indication in the figure does not meet the ±. It is only +. Same for Fig. 5B also.
Response 11: We improved the description of all the figures in the manuscript, and made modifications to the legends of Figures 4, 5, and 6 as suggested. Thank you for your correction.
Comments 12: Figure 5A. WT, L1, L2, L3 indication in the fig. is not appropriate. Point correctly. Either at upper or lower of the fig.
Response 12: We revised the Figure 5 as suggested. Thanks for your useful suggestion.
Comments 13: This is somewhat weak discussion for the journal Plants. I suggest, some improvement having results comparing with recent article in ramie as well as Arabidopsis, even other plants related to WRKY functions, structures, and others.
Response 13: We rewrote the Discussion. Thanks for your useful suggestion.
Comments 14: -need some discussion about cultivation
Responsen 14: Revised as suggested. Thanks for your useful suggestion.
Comments 15: -It is important to indicate the time of database browsed
Response 15: Revised as suggested. Thanks for your useful suggestion.
Comments 16: L272; Total RNA was extracted from where?
Response 16: The total RNA was extracted from the samples in 4.1, include the samples of roots and leaves under drought and salt stress. and the samples from different growth stage (the seedling stage, the rapid-growth stage and the fiber maturation stage), leaves and stem bark (bottom, middle and top parts). We rewrite the 4.2. to help readers understand. Thanks for your useful suggestion.
Comments 17: L328; Leaves from where?
Response 17: We rewrote the 4.8. to help readers understand. Thanks for your useful suggestion.
Comments 18: -Enzymatic assay need description of references
Response 18: The determination methods for the three enzymes were based on the instructions of the reagent kit method, and we added two references for POD and CAT according to the instructions. Thanks for your useful suggestion.
Comments 19: Most analysis was from Arabidopsis, not from ramie
Response 19: We rewrote the part of the Discussion and added the analysis from ramie. Thanks for your useful suggestion.
Comments 20: Conclusion is too scanty. Does not represent the title and abstract. However, have reflection of partial results.
Response 20: We rewrote the Conclusion as as suggested. Thanks for your useful suggestion.
Thank you very much for your comments and suggestions. We hope that the revision is acceptable, and I look forward to hearing from you soon.
Sincerely,
Yaning Bao
Round 2
Reviewer 1 Report
Comments and Suggestions for Authors
The Ms has been improved but it still needs corrections.
Lines 70-75 is not properly cited. ref 13-14 are wrongly cited. several studies reported the role of SOD, CAT, GPX, MDHAR, GR, AAO etc. in various plants including the crop plant bread wheat, and even TaRBOH during drought stress. Authors should properly discuss this and cite proper relevant references. These studies might also be discussed in the discussion where authors showed the antioxidant enzyme activities, because most of the above-suggested studies in bread wheat is directly linked with the drought and other stresses and showed the gene expression change or enzyme activity.
Besides, several other genes including antoxidants encoding and transporters have been used to achieve the drought tolerance in transgenic Arabidopsis. A few latest reports like the expression of TaGPX1-D and TaNCL2-A could be the part of introduction as well as discussion.
Figure quality should be further improved. Scales are not visible in the figures.
Comments on the Quality of English LanguageModerate editing of English language required
Author Response
Dear Reviewer,
Thank you very much for taking the time to review this manuscript entitled “Molecular Clone and expression analysis of the WRKY transcription factor family in ramie (Boehmaeria nivea L. Gaud) and Overexpression of BnWRKY49 increases drought tolerance in Arabidopsis” (Manuscript Number: plants-2830052). Thank you for your rigorous consideration, we totally understand the reviewer’s concern.
Comments 1: Lines 70-75 is not properly cited. ref 13-14 are wrongly cited. several studies reported the role of SOD, CAT, GPX, MDHAR, GR, AAO etc. in various plants including the crop plant bread wheat, and even TaRBOH during drought stress. Authors should properly discuss this and cite proper relevant references. These studies might also be discussed in the discussion where authors showed the antioxidant enzyme activities, because most of the above-suggested studies in bread wheat is directly linked with the drought and other stresses and showed the gene expression change or enzyme activity.
Besides, several other genes including antoxidants encoding and transporters have been used to achieve the drought tolerance in transgenic Arabidopsis. A few latest reports like the expression of TaGPX1-D and TaNCL2-A could be the part of introduction as well as discussion.
Response 1: We rewrote the L70-75 as suggested. We deleted the ref.13-14, and added the related latest reports of ref.13, 16, and 39 in the introduction or discussion as suggested. Thanks for your useful suggestion.
Reference:
[13] Sharma, Y.; Ishu; Shumayla; Dixit, S.; Singh, K.; Upadhyay, S.K. Decoding the features and potential roles of respiratory burst oxidase homologs in bread wheat. Curr. Plant Biol. 2024, 37, 100315.
[16] Tyagi, S.; Shumayla; Sharma, Y.; Madhu; Sharma, A.; Pandey, A.; Singh, K.; Upadhyay, S.K. TaGPX1-D overexpression provides salinity and osmotic stress tolerance in Arabidopsis. Plant Sci. 2023, 337,111881.
[19] Shumayla; Tyagi, S.; Sharma, Y.; Madhu; Sharma, A.; Pandey, A.; Singh, K.; Upadhyay, S.K. Expression of TaNCL2-A ameliorates cadmium toxicity by increasing calcium and enzymatic antioxidants activities in Arabidopsis.Chemosphere 2023, 329, 138636.
Comments 2: Figure quality should be further improved. Scales are not visible in the figures.
Response 2: We replaced all the images with better quality. In addition, we uploaded the original image to the Submission system. Thanks for your useful suggestion.
Thank you very much for your comments and suggestions. We hope the revision is acceptable, and I look forward to hearing from you soon.
Sincerely,
Yaning Bao
Round 3
Reviewer 1 Report
Comments and Suggestions for Authors
Ms may now be accepted.
Comments on the Quality of English LanguageMinor changes required
Author Response
Dear Reviewer,
Thank you very much for taking the time to review this manuscript entitled “Molecular Clone and expression analysis of the WRKY transcription factor family in ramie (Boehmaeria nivea L. Gaud) and Overexpression of BnWRKY49 increases drought tolerance in Arabidopsis” (Manuscript Number: plants-2830052). Thank you very much for your comments and suggestions.
Our manuscript has been edited with appropriate English language, grammar, punctuation, spelling, and overall style on the AJE website. The verification code was 6F23-C861-7A84-FEF1-6C6A. Before this submission, we revised the manuscript's content, so we have carefully checked and improved the English writing in the revised manuscript. We hope the revision is acceptable, and I look forward to hearing from you soon.
Sincerely,
Yaning Bao